# JAUNE: JUSTIFIED AND UNIFIED NEURAL LANGUAGE EVALUATION

## ABSTRACT

We review the limitations of BLEU and ROUGE – the most popular metrics used to assess reference summaries against hypothesis summaries, and introduce JAUNE: a set of criteria for what a good metric should behave like and propose concrete ways to use recent Transformers-based Language Models to assess reference summaries/translations against hypothesis summaries/translations.

## 1 INTRODUCTION

Evaluation metrics play a central role in the machine learning community. They direct research efforts and define the state of the art models. In machine translation and summarization, the two most common metrics used for evaluating similarity between candidate and reference texts are BLEU (Papineni et al., 2002) and ROUGE (Lin, 2004). Both approaches rely on counting the matching n-grams in the candidate text to n-grams in the reference text. BLEU is precision focused while ROUGE is recall focused.

These metrics have posed serious limitations and have already been criticized by the academic community (Reiter, 2018) (Callison-Burch et al., 2006) (Sulem et al., 2018) (Novikova et al., 2017). In this work, we formulate an empirical criticism of BLEU and ROUGE, establish JAUNE: a set of criteria that a sound evaluation metric should pass. Furthermore we propose concrete ways to use recent advances in NLP to design data-driven metrics addressing the weaknesses found in BLEU and ROUGE while scoring high on the criteria for a sound evaluation metric.

## 2 RELATED WORK

### 2.1 BLEU, ROUGE AND N-GRAM MATCHING APPROACHES

BLEU (Bilingual Evaluation Understudy) (Papineni et al., 2002) and ROUGE (Recall-Oriented Understudy for Gisting Evaluation) (Lin, 2004) have been used to evaluate many NLP tasks for almost two decades. The general acceptance of these methods depend on many factors including their simplicity and intuitive interpretability. Moreover, the main factor is the claim that they highly correlate with human judgement (Papineni et al., 2002).

The shortcomings of these methods have been widely criticised and studied. Reiter (Reiter, 2018), in his structured review of BLEU, finds a low correlation between BLEU and human judgment. Callison et al (Callison-Burch et al., 2006) examine BLEU in the context of machine translation and find that BLEU neither correlates with human judgment on adequacy (whether the hypothesis sentence adequately captures the meaning of the reference sentence) nor on fluency(the quality of language in a sentence). Sulem et al (Sulem et al., 2018) examine BLEU – in the context of text

simplification – on grammaticality, meaning preservation and simplicity. They report a very low, and, in some cases, negative correlation with human judgment.

Considering these results, it is a natural step to pursue new avenues for natural language evaluation and, with the advent of deep learning, using neural networks for this task is a promising step forward.

## 2.2 TRANSFORMERS, BERT AND GPT

Language modeling has become an important NLP technique, thanks to the ability to apply it to various NLP tasks as explained in Radford et al (Radford et al., 2019). There are two leading architectures for language modeling: Recurrent Neural Networks (RNNs)(Mikolov et al., 2010) and Transformers (Vaswani et al., 2017). RNNs handle the input tokens, words or characters, one by one through time to learn the relationship between them, whereas, transformers receive a segment of tokens and learn the dependencies between them using an attention mechanism.

## 2.3 MODEL-BASED METRICS

While BLEU and ROUGE are defined in a discrete space, new evaluation metric can be defined in this continuous space. BERTscore (Zhang et al., 2019) uses word embeddings and cosine similarity to create a score array and uses greedy matching to maximize the similarity score. Sentence Mover's Similarity (Clark et al., 2019) uses the mover similarity, Wasserstein distance, between sentence embedding generated from averaging the word embeddings in a sentence.

Both of these methods report stronger correlations with human judgment and better results when compared to BLEU and ROUGE. While they are using word embeddings (Mikolov et al., 2013) to transfer their sentence in a continuous space, they are still using distance metrics to evaluate that sentence. BLEND (Ma et al., 2017) uses an SVM to combine different existing evaluation metrics.

Another proposed evaluation method is RUSE (Shimanaka et al., 2018). This method embeds both sentences separately and pool them to a given size. After, the method uses a pre-trained MLP to predict on different tasks. This quality estimator metric is then proposed to be used in language evaluation.

Our proposed methodology is to take neural language evaluation beyond architecture specifications. We are proposing a framework in which an evaluator's success can be determined.

## 2.4 GLUE BENCHMARK

The GLUE Benchmark is a tool for evaluating and analyzing the performance of models across a diverse range of existing NLU tasks (Wang et al., 2018). The recent introduction of this benchmark has catalyzed the development of architectures scoring well on a wide variety of tasks and encouraged the NLP community to move away from specialized models doing well on a single task to models performing well across benchmarks. The variety of tasks introduced in the GLUE Benchmark are linguistic acceptability, sentiment analysis, semantic similarity, question answering, logical inference and reading comprehension. To be assessed according to that benchmark, models such as Transformers are usually pre-trained on a large corpus in an unsupervised manner and fine-tuned on a dataset used for the specific task of the benchmark.

## 3 CHALLENGES WITH BLEU AND ROUGE

In this part, we will discuss the limitations of BLEU and ROUGE. There are simple ways to attack these n-gram based metrics like adding a single word negation or changing all possible words with synonyms. Although these are theoretically plausible scenarios we also wanted to analyze which cases forced these metrics to fail in real life.

We took 100 examples from the STS-B dataset (Cer et al., 2017) where the absolute difference between the BLEU/ROUGE score and normalized label was the biggest. This does not necessarily capture all failure cases of BLEU/ROUGE, but a variety of failure cases can be observed. Through this analysis, we see that there are systematically recurring real life examples, just like in our theo-

retical examples, where BLEU and ROUGE are failing to assess the level of similarity between two sentences.

We also observe that though some of the shortcomings of unigram metrics are mitigated through higher order n-grams, they open the door for different problems. Some of the most common failure cases that we have encountered are listed in table 1.

## 3.1 IDIOMS AND ADDING DETAILS

A commonly encountered failure mode involves the use of idioms and the addition of extra examples/details in one of the sentences. These types of errors are especially common in more natural conversations. These types of errors also made 25 % of our analysis. Here, we characteristically see humans giving high scores to these sentences because they are aware of which part holds the core meaning of the sentence while BLEU/ROUGE lack this ability.

An example from the dataset is "You should take this animal to a vet right away." and "As covered in the other answers, your only option is to see a vet in order to have surgery done." while the true score for this sentence pair is 3.6 out of 5, BLEU gives this sentence a 0.46 out of 5.

## 3.2 CHANGING WORDS

Another common failure mode of BLEU and ROUGE occurs when one or a few important words of a sentence is changed while the rest of the structure is kept the same.

An example from STS-B is "a man is speaking." and "a man is spitting.". While human judges give these two sentences a similarity score of 0.64 out of 5, BLEU-1 gives this sentence a 3.75 out of 5.

## 3.3 GENERAL PARAPHRASE

We frequently see BLEU and ROUGE failing in the context of:

- General paraphrases where words are replaced by synonyms. This accounts for 12 % of cases.

- Sentences are reordered. This accounts for around 10% of observed errors.

- Different verb tenses are used. This accounts for 15% of observed errors.

While the higher order n-grams are supposed to preserve the intelligibility of the sentence and not reward a model that outputs words in a random order, they also punish valid re-orderings of sub sentences or words. In these smoothed methods, changing a word with a synonym will also result in a much higher penalty.

BLEU and ROUGE are methods that are much more frequently under scoring sentence pairs than over scoring them. Similarly to Reiter (Reiter, 2018) we conclude that BLEU/ROUGE can be fruitful in deciding whether a model is bad but not whether it is good.

We also provide detailed examples of these failure cases in the appendix.

## 3.4 EXPERIMENTS

### 3.4.1 SOME EXAMPLES FROM STS-B SENTENCES PAIRS

To illustrate our argument, we will give some examples from the dataset with their BLEU/ROUGE scores as well as a score generated from a RoBERTa model fine tuned on the STS-B dataset. We refer to this model as RoBERTa-STS. Note that in this paper the BLEU* and ROUGE* scores are not between 0 and 1 but are scaled with 5 to be more understandable with the scale of the scoring metric used in the dataset. That is why we refer to them as BLEU* and ROUGE*

Table 1: BLEU*/ROUGE* and RoBERTa-STS scores on sentence pair examples from STS-B

| Sentence pair | BLEU* | ROUGE* | RoBERTa-STS | Label |
|---|---|---|---|---|
| The last time the survey was conducted, in 1995, those numbers matched.
In 1995, the last survey, those numbers were equal. | 0.99 | 1.42 | **4.65/5** | **5.00/5** |
| A band is performing on a stage.
A band is playing onstage. | 1.14 | 2.29 | **3.85/5** | **5.00/5** |
| Two white dogs are swimming in the water.
The birds are swimming in the water. | 3.00 | 3.23 | **1.19/5** | **0.80/5** |
| A man plays the piano.
A man is playing a piano. | 0.92 | 2.17 | **5.00/5** | **5.00/5** |
| Pardon the brevity of this answer, but I would say "named" is preferred within the context of your example.
Named is preferred in your example, since you are formally giving a name to your method. | 0.25 | 0.72 | **3.73/5** | **4.40/5** |

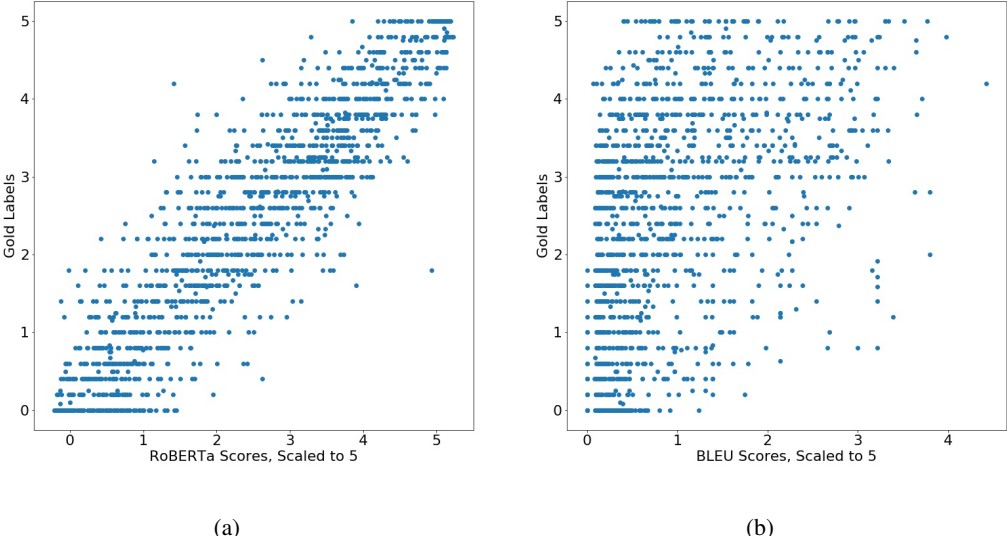

(a)                                                    (b)

Figure 1: Comparison of RoBERTa-STS scores(a) and BLEU* scores(b) with labels from the STS-B dev set.

### 3.4.2  SEMANTIC SIMILARITY EXPERIMENTS

In figure 1 we can see that development set scores of RoBERTa-STS in figure 1a and BLEU* in figure 1b compared to the gold labels. While we will look at the correlation scores, we can also see that the average errors in BLEU and RoBERTa-STS are remarkably different.

Table 2: Correlation with human judgement of similarity on STS-B Benchmark development set

| | ROUGE | BLEU | RoBERTa-STS |
|---|---|---|---|
| Pearson correlation with human judgement | 0.55 | 0.50 | **0.92** |

|  | ROUGE | BLEU | RoBERTa-STS |
|---|---|---|---|
| Spearman's RC | 0.255 | 0.216 | **0.744** |
| Kendall's $\tau$ | 0.215 | 0.186 | **0.69** |

Table 3: Results of logical entailment experiments

# 4 ASSESSING EVALUATION METRICS

## 4.1 METRIC SCORECARD

To overcome the previously highlighted challenges and provide a framework in which metrics comparing reference summaries/translation can be assessed and improved, we establish first-principles criteria on what a good evaluator should do:

- The first one is that it should be highly correlated with human judgement of semantic similarity.

- The second one is that it should be able to distinguish sentences which are in logical contradiction, logically unrelated or in logical agreement.

- The third one is that given s1, s2 which are semantically similar, eval(s1,s2) $\geq$ eval(s1,s2(corrupted) $\geq$ eval(s1,s2(more corrupted)) where corruption here includes removing words, adding noise to the word order or including grammatical mistakes.

## 4.2 IMPLEMENTING THE METRIC SCORECARD

We will now show how the scorecard can be implemented. For every dimension of the scorecard, the experiments are done with three metrics. BLEU with equal weights between 1 to 4 grams. ROUGE with averaging ROUGE-1 and ROUGE-2 and a neural evaluator. The evaluator is the RoBERTa-STS.

### 4.2.1 SEMANTIC SIMILARITY

The first expectation from a good similarity metric is to correlate highly with human judgment in terms of assessing semantic similarity. Here we assessed BLEU and ROUGE on the STS-B benchmark and compared their performance to a RoBERTa model fine tuned for semantic similarity (Table 2).

### 4.2.2 LOGICAL ENTAILMENT

Another characteristic of a good metric is to differentiate the argument, core meaning in a sentence and take it into account when assessing hypothesis text with references. Here, we used the MNLI dataset where ,for each text, we have three hypothesis texts representing contradiction, neutral and entailment. We expect a good metric to rank entailment higher than neutral, and ,both of them, higher than contradiction. To assess the quality of a metric, we propose to use the Spearman's ranked correlation and ,in (Table 3), we also experiment with Kendall's $\tau$. Here, we observe that the RoBERTa-STS model remarkably outperforms BLEU and ROUGE and both of these metrics show very little correlation with human judgment.

### 4.2.3 ROBUSTNESS TO GRAMMATICAL ERRORS

For assessing the third criteria. We start with 3479 sentence pairs from the MNLI dataset that are labelled as entailment. We introduce random corruptions such as random insertion, deletion and grammatical errors as in (Zhao et al., 2019). We use two different set of parameters for different corruption levels, and expect that a good metric would rank the original similar sentence higher than the less corrupted and both higher than the more corrupted sentence. Here, we also propose to use the Spearman's ranked correlation and also experiment with Kendall's $\tau$. We report results on Table 4. Where we see that the RoBERTa-STS model once more outperforms BLEU and ROUGE.

|  | ROUGE | BLEU | RoBERTa-STS |
|---|---|---|---|
| Spearman's RC | 0.528 | 0.472 | **0.718** |
| Kendall's $\tau$ | 0.478 | 0.419 | **0.667** |

Table 4: Results of grammatical error experiments

|  | ROUGE-1 | ROUGE-2 | ROUGE-L | BLEU | RoBERTa-STS |
|---|---|---|---|---|---|
| Pearson Corr. | 0.498 | 0.491 | 0.526 | 0.253 | **0.63** |

Table 5: Results of WMT experiments

## 5 MACHINE TRANSLATION EXPERIMENTS

In the previous section, we outlined a series of criteria to assess evaluation metrics and shown how, for each dimension, RoBERTa-STS significantly outperformed ROUGE and BLEU. In the following section, we report results showing how BLEU, ROUGE and RoBERTa-STS correlate with human judgement of quality in the case of machine translation.

Using the WMT2015 and WMT2016 datasets, we collected reference and hypothesis translations ,along with human judgment scores, where the translation was from another language to English. This gave us 5360 sentence pairs. For each of these sentence pairs, we computed the BLEU, ROUGE-1, ROUGE-2, ROUGE-L scores. We also computed the RoBERTa-STS predicted semantic similarity.

We computed the correlation of the various scores with human judgements and reported the scores in Table 5. Here again, we can see Transformers based model outperforming BLEU and ROUGE.

## 6 CONCLUSION AND FUTURE WORK

In this work, we have established a framework to assess metrics comparing the quality of reference and hypothesis summary/translations. Based on these criteria, we compare evaluators using recent Transformers (in this case RoBERTa-STS) to BLEU and ROUGE. We also show how this good performance on our scorecard translates on a previously unseen machine translation datasets. Such results highlight the potential to replace BLEU and ROUGE with data-driven models such as RoBERTa-STS.

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

## A APPENDIX

In the appendix, we will discuss the failure cases of BLEU, ROUGE and RoBERTa-STS in detail to provide a better understanding of how these models can fall short in language evaluation. This is important because a good metric scorecard has to represent the quality of an evaluator. These experiments are to show that our metrics cover many of the failure cases and can assess them without the burden of manually evaluating the outputs of every evaluator.

We will start by taking examples from the BLEU and ROUGE dataset. As in the paper the BLEU scores used are always a uniform average up to 4-grams and the ROUGE score is the average of ROUGE-1 and ROUGE-2. Both scores are scaled up to 5 to increase the interpretebility of the scores given that the labels in the similarty dataset are between 0 and 5.

Table 6: BLEU*/ROUGE* and RoBERTa-STS scores on sentence pair examples from STS-B

| Id | Sentence pair | BLEU* | ROUGE* | RoBERTa-STS | Label |
|---|---|---|---|---|---|
| 1 | The company claims it's the largest single Apple VAR Xserve sale to date. The company claimed it is the largest sale of Xserves by an Apple retailer. | 0.44 | 1.62 | **4.36/5** | **5.00/5** |

| 2 | A woman puts flour on a piece of meat.
A woman is putting flour onto some meat. | 0.63 | 1.78 | **5.07/5** | **5.00/5** |
|---|---|---|---|---|---|
| 3 | He later learned that the incident was caused by the Concorde's sonic boom.
He later found out the alarming incident had been caused by Concorde's powerful sonic boom. | 0.74 | 2.58 | **4.96/5** | **5.00/5** |
| 4 | It indeed appears the Andromeda galaxy (M31) and The Milky Way (MW) are en route to a collision.
In a few billion years, the Milky Way and Andromeda will collide. | 0.20 | 1.09 | **3.37/5** | **4.40/5** |
| 5 | You definitely do NOT want to be supporting your weight with your arms on the bike for normal riding.
No, don't support your weight on your arms Your hands simply aren't really made for supporting all that weight. | 0.28 | 1.13 | **2.73/5** | **4.20/5** |
| 6 | 7 detained for 'house sister' scandal
China detains 7 for "house sister" scandal | 0.33 | 1.52 | **4.25/5** | **4.20/5** |
| 7 | A man plays the violin.
A man is playing violin. | 1.14 | 2.41 | **5.12/5** | **5.00/5** |
| 8 | It is simply the number of balls bowled divided by the number of wickets taken.
Bowling strike rate is defined for a bowler as the average number of balls bowled per wicket taken. | 0.80 | 1.84 | **3.83/5** | **4.40/5** |
| 9 | Police helicopter crashes into pub in Glasgow - several casualties
Helicopter crashes into roof of Glasgow club | 0.47 | 1.36 | **3.58/5** | **4.00/5** |
| 10 | Oil falls in Asian trade
Oil prices down in Asian trade | 1.62 | 3.14 | **4.89/5** | **5.00/5** |
| 11 | A skateboarder jumps off the stairs.
A dog jumps off the stairs. | 3.21 | 3.77 | **1.09/5** | **0.80/5** |
| 12 | Wigan 3-2 Wolves: Match report, pictures & video highlights
Arsenal 0-0 Chelsea: Match report, pictures & video highlights | 3.39 | 3.26 | **0.58/5** | **1.20/5** |

In table 6 we see examples of many different error cases and ,in most sentences, we also have more than one cause for the drastic difference between BLEU/ROUGE and the label. For instance, in rows 1 and 6 we see that the cause for the error is the reordering of sub-sentences, spelling/punctuation and newly introduced words that don't change the meaning but merely extend it. While BLEU and ROUGE are failing in these examples, we see that the RoBERTa-STS model scores similarly to the label. In line 7, we can see that the RoBERTa-STS model score is above 5.

In rows 2 and 7, we see that the main difference is the form or tense of the verb in a sentence. This makes BLEU severely under score simple changes with synonyms or valid re-orderings as seen in the examples below. This characteristic of BLEU reinforces the point that BLEU and ROUGE are not useful in tracking the state of the art and comparing the best methods but are tools to weed out bad models fairly simply.

In rows 3 and 9 we see sentences that differ due to using descriptive phrases instead of a word or extending the sentence with more information. These types of errors changes are also caught with language models since we know they have the ability to hold the meaning of multiple words and incorporate them to reach a related word as in the famous example of king - men + woman = queen Mikolov et al. (2013).

In rows 4,8 and 5 we see general paraphrases with the same meaning represented in a generally different sentence. In all cases we see a drastic difference between BLEU/ROUGE and the label but these cases also unearth a specific characteristic of the neural evaluator. In 4 and 8 we see that the

error of the RoBERTa-STS model comparatively lower than row 5. While it is hard to determine the exact cause through only looking at these examples table 7 for the RoBERTa-STS failure cases will make this case more compelling.

While language models have a general sense of the context in a given sentence, they still lack a general knowledge of the world. Hence in the second sentence of row 5, because the words riding, bike, bicycle are missing the model has a hard time recognising that the second sentence is also about the same topic. To test this we added "while riding" or "on a bike" at the end of a sentence and the score immediately went up to 3.6/5 while barely changing the BLEU* and the ROUGE score. In row 4 and 8 however, the context of the sentence is defined explicitly with the key phrases. We see this bias affecting RoBERTa-STS scoring in the examples below.

Table 7: BLEU*/ROUGE* and RoBERTa-STS scores on sentence pair examples from STS-B

| Id | Sentence pair | BLEU* | ROUGE* | RoBERTa-STS | Label |
|---|---|---|---|---|---|
| 1 | It would be unusual for a snake to attack a stationary person. I'm no herpetologist, but in my experience, snakes are in the "you don't bug me, I won't bug you" category. | 0.34 | 0.00 | **1.4/5** | **4.20/5** |
| 2 | New UN peacekeeping chief named for Central African Republic UN takes over peacekeeping in Central African Republic | 1.16 | **2.39** | 3.69/5 | **2.00/5** |
| 3 | From Broadway comedies like "The Seven Year Itch" (1952), "Will Success Spoil Rock Hunter?" Playwright George Axelrod, who anticipated the sexual revolution with The Seven Year Itch and Will Success Spoil Rock Hunter? | **2.03** | 1.31 | 3.16/5 | **2.00/5** |
| 4 | a group of navy seals are singing A group of military personnel are playing in a brass quintet. | 0.40 | **1.45** | 0.75/5 | **2.40/5** |

In the above examples, we will find two points that will helps us better understand the RoBERTa-STS as a neural evaluator. Firstly, we see that the neural network sometimes lacks a sense of context that is not given in the sentence explicitly. While these language models are trained on a large corpus and capture a sense of the words and language, we still see that their performance is not perfect. We see these examples in row 4, where the model cannot relate a navy seal as a military personnel. Or as in row 1, where the model cannot model an idiom.

The second and more critical place where we need further development is especially detecting whether the core argument/message in a sentence is the same beyond whether if they are talking about the same things. As in rows 2 and 3. We see the same landmark words and can clearly say that the sentences are talking about the same things, but what a human can distinguish is that they are saying unrelated things. This is one of the key motivations in including the language inference task in the scorecard. Since detecting whether a pair of sentences are related on what level is a key part of detecting sentence similarity.

One last thing we will mention is that while RoBERTa-STS and BLEU/ROUGE have different error cases, their performance on these error cases is also remarkably different in favor of the former. Table 8 shows the mean error of BLEU* and the RoBERTa-STS model on each others top 500, which is one third of the development set, error cases.

Table 8: Average error of BLEU* and RoBERTa-STS in the their low scoring sets. With rows corresponding to which models failure cases and the columns to which model is used to score

| | BLEU* | RoBERTa-STS |
|---|---|---|
| BLEU* | 2.93/5 | **0.47/5** |
| RoBERTa-STS | 1.68/5 | **0.89/5** |

We see in table 8 that BLEU* has a remarkable error in both its failure cases and also the failure cases of RoBERTa-STS while RoBERTa-STS outperforms BLEU* in each category.

While neural evaluators have also room for improvement, we can with confidence say that they are outperforming classical methods and with a methodical way of improving them can bolster progress of NLP research.

