# OpenReview forum: "JAUNE: Justified And Unified Neural language Evaluation"
_ICLR.cc/2020/Conference — Reject_

### Official Review · AnonReviewer2 · 2019-10-22
**Official Blind Review #2**

**Rating:** 1

**Review:**

This paper argues that BLEU and ROUGE, two metrics that are used for the evaluation of machine translation and text summarization systems, are flawed, and proposes a new JAUNE metric to replace it.

The authors train a regressor on the STS-B dataset, and show that their model (which is using sentence embeddings from RoBERTa) corresponds better to the ground truth similarity labels than then scaled (but otherwise unchanged) BLEU scores. This is probably not surprising, given the small size and specific nature of the STS-B task and dataset.

I could agree with many of the problems that the authors describe, but the proposed solution seems to be a very specific solution that works on a given dataset (for which supervised training data is available), but I do not think it will generalize well to unseen test data in different domains. I also do not understand how the BLEU score can simply be rescaled from 0-5 - how do you determine the maximum BLEU score before rescaling?

The paper should be proofread by a native speaker, many sentences are unclear, and spacing as well as punctuation are used in weird ways.


**Experience Assessment:**

I have read many papers in this area.

**Review Assessment: Checking Correctness Of Derivations And Theory:**

I did not assess the derivations or theory.

**Review Assessment: Checking Correctness Of Experiments:**

I assessed the sensibility of the experiments.

**Review Assessment: Thoroughness In Paper Reading:**

I read the paper at least twice and used my best judgement in assessing the paper.

---

> ### Author Response · Authors · 2019-11-15
> **Revision includes new experimental results on WMT showing RoBERTa STS-regressor still outperforms ROUGE/BLEU**
>
> Dear Reviewer #2,
>
> Thank you for sharing your concerns around the ability of the RoBERTa-STS regressor to generalize beyond the STS-B dataset. We have included experimental results performed on WMT. The trends are the same.
>
> It is also to be noted that the regressor is less specialized than you make it appear. Firstly, the starting point of the regressor is a BERT model. In "BERT Rediscovers the Classical NLP Pipeline", it is shown that these general language models are unsupervised multitask learners.
>
> The goal of fine-tuning on STS-B was to train the regressor to predict semantic similarity. The WMT experiments show that the model outperforms BLEU/ROUGE by 20+% on correlating with human judgment for 5k+ WMT sentence pairs.
>
> As explained in section 3.4.1 "BLEU* and ROUGE* scores are not between 0 and 1 but are scaled with 5 to be more understandable with the scale of the scoring metric used in the dataset (STS-B)"

---

### Official Review · AnonReviewer3 · 2019-10-25
**Official Blind Review #3**

**Rating:** 1

**Review:**


=== Summary ===

The authors motivate the development of new (automatic) metrics to evaluate language generation by using similarity with a given reference: standard metrics like BLEU, ROUGE or METEOR have been shown to have poor correlation with human judgment on a number of tasks and are vulnerable to changes in word re-ordering, semantics-changing word replacement, and syntactic transformations.

They then propose a multi-dimensional evaluation criteria to evaluate sentence similarity based on semantic similarity (something that correlates with human judgments of the same), logical equivalence and fluency.

The paper then goes on to describe possible directions to tackle several key problems in evaluation: evaluating semantic similarity by using models trained on the GLUE benchmark, evaluating logical equivalence using models trained on the MNLI corpus and fluency based on the CoLA corpus.

=== Decision ===

The problem this paper seeks to tackle is clearly one of great
importance in the field, but I find it hard to argue that this paper
significantly contributes to the existing body of work (more on this
below) and as a result I vote to reject this paper.

There are two possible contributions for this paper: a set of criteria for what makes a good evaluation metric and the concrete proposals to implement these criteria.

For the first, I find the proposed criteria to be overly generic and not helpful at providing additional clarity on what makes for a good evaluation: for example, how is semantic similarity different from logical consistency? Does it make sense to compare the semantic similarity of two sentences if one of them isn't even near grammatical? A lot of prior work already argue the shortcomings of the existing metrics this paper is making, e.g. Conroy and Dang (2008), Liu et al. (2016), Novikova et al. (2017). I think it would be valuable to present new axes to decompose the evaluation problem, but more work is needed to clarify and develop the axes presented in this paper.

For the second possible contribution, the idea of evaluating language generation along dimensions is not novel and in fact quite standard in the NLP community. The challenge has been showing that there are subset of tasks that can be used a reliable metrics across different domains and systems. Unfortunately, this paper does not actually evaluate its own proposals, making it hard to evaluate how effective its proposals are.


**Experience Assessment:**

I have published in this field for several years.

**Review Assessment: Checking Correctness Of Derivations And Theory:**

N/A

**Review Assessment: Checking Correctness Of Experiments:**

N/A

**Review Assessment: Thoroughness In Paper Reading:**

I read the paper at least twice and used my best judgement in assessing the paper.

---

> ### Author Response · Authors · 2019-11-15
> **Revision focuses on improving the scorecard with additional experimental results and tests proposed metric**
>
> Dear Reviewer #3,
>
> Thank you for your comments. We have evaluated the proposed metric according to the scorecard showing it performs better on the proposed scorecard than BLEU/ROUGE.
>
> We further show how this scorecard performance translates into a similar performance on WMT sentence pairs.
>
> Finally, the scorecard has been further fleshed out and is now actionable enough to assess any metric comparing candidate text with reference text.

---

### Official Review · AnonReviewer1 · 2019-10-27
**Official Blind Review #1**

**Rating:** 1

**Review:**

Observing shortcomings of BLEU and ROUGE, the paper proposes, JAUNE, a set of criteria for a good evaluation metric. These criteria include: high correlation with human judgement; being able to distinguish similar but contradicting statements; penalizing grammatical errors, and hard to game.

The paper, as its current form, is not ready for publishing. Some suggestions and comments:

- Please carefully check the paper and fix typos and confusing sentences. I was collecting these errors but eventually stopped. Some examples. Sec. 2.3: punctuation missing between "RUSE" and "this method", comma missing after "a discrete space"; Sec. 4.1.1: "made to ,for example"....

- The motivation of the paper is unclear. Is your criticism only about BLEU and ROUGE, or the state of the arts in NLP evaluation in general? To make JAUNE appealing, one has to argue that the state of the arts in NLP evaluation is ineffective. For this, the paper needs to review a boarder range of metrics beyond just BLUE and ROUGE.

- While the authors suggest a data-driven metric, it reads to me like a model-driven metric (RoBERTAa specifically). Doesn't it systematically bias towards a certain family of metrics?

- Better and more comprehensive experimental results are highly desired.

**Experience Assessment:**

I have read many papers in this area.

**Review Assessment: Checking Correctness Of Derivations And Theory:**

N/A

**Review Assessment: Checking Correctness Of Experiments:**

I assessed the sensibility of the experiments.

**Review Assessment: Thoroughness In Paper Reading:**

N/A

---

> ### Author Response · Authors · 2019-11-15
> **Revision includes more comprehensive experimental results and addresses comments above**
>
> Dear Reviewer #1,
>
> 1) Thank you for your comments. We have checked the paper for typos, confusing sentences and made it easier to read.
>
> 2) Overall the goals of the paper are to present clear criteria to assess evaluation metrics and show how transformers can be used to assess the quality of candidate translations and summaries.
>
> 3) Currently, all summaries/translation results use BLEU/ROUGE, hence the focus on these 2 metrics. The core idea in the experimental section is that in all dimensions capturing the properties of a "good evaluator", BLEU/ROUGE are outperformed by a Transformer trained to predict semantic similarity between candidate and reference sentences. The earlier version of the paper hinted at a more general framework in which Transformers themselves are used as feature extractors for various sentence pairs but we removed this part to reduce confusion.
>
> 4) We use the term data-driven because the evaluators are general purpose language models fine-tuned on semantic similarity. We took RoBERTa-STS as an example but recent, high-performing models such as ALBERT or ALICe are also applicable. The constant among those is the training procedure relying on the same data. The models are anticipated to evolve which is why we did not want to be specifying just a model.
>
> 5) We have added 3 more experiments. The first 2 complete the evaluation of RoBERTa-STS for the 2 dimensions of the scorecard. The last experiments use RoBERTa-STS on 5300+ WMT sentence pairs and show that this model transfers well.

---

### Decision · Program_Chairs · 2019-12-19

**Decision:**

Reject

**Comment:**

The authors tackle the questions of automatic metrics for assessing document similarity and propose the use of Transformed-based language models as a critic providing scores to samples. As a note, ideas like these have been also adopted in Computer Vision with the use of the Inception score as a proxy the quality of generated images. The authors ask great questions in the paper and they clearly tackle a very important problem, that of automatic measures for assessing text quality. While their first indications are not negative, this paper lacks the rigor and depth of experiments of a conference paper that would convince the research community to abandon BLEU and ROUGE in lieu of some other metric. It's perhaps a good workshop paper or a short paper at a *CL conference. Specifically, we would need more tasks where BLEU/ROUGE is the standard measure and so how the proposed measure correlates better with humans,  so cases where word overlap is in theory a good proxy of similarity assuming reference sentence (e.g., logical entailment is not such a prototypical task). MT is a first step towards that, but summarization is also a necessary I would say. Other questions of interest relate to the type of LM (does it only need to be Roberta?) and the quality of LM (what if i badly tune my LM?)  On a more personal note: We all know that BLEU is not a good metric (especially for document-level judgements) and every now and then there have been proposals to replace BLEU that do correlate better (e.g., http://ccc.inaoep.mx/~villasen/bib/Regression%20for%20machine%20translation%20evaluation.pdf) . However, BLEU is still here due to each simplicity. Please keep pushing this research and I’m looking forward to seeing more experimental evidence.